# MagiC: Evaluating Multimodal Cognition Toward Grounded Visual Reasoning

## Abstract

Recent advances in large vision-language models have led to impressive performance in visual question answering and multimodal reasoning. However, it remains unclear whether these models genuinely perform grounded visual reasoning or rely on superficial patterns and dataset biases. In this work, we introduce **MagiC**, a comprehensive benchmark designed to evaluate grounded multimodal cognition—assessing not only answer accuracy but also the quality of step-by-step reasoning and its alignment with relevant visual evidence. Our benchmark includes 5,534 weakly supervised QA examples generated from strong model outputs and 896 human-curated examples with fine-grained annotations, including answers, rationales, and bounding box groundings. We evaluate 15 vision-language models ranging from 7B to 70B+ parameters across four dimensions: final answer correctness, reasoning validity, grounding fidelity, and self-correction ability. **MagiC** further includes diagnostic settings to probe model robustness under adversarial visual cues and assess their capacity for introspective error correction. We introduce new metrics such as *MagiScore* and *StepSense*, and provide comprehensive analyses that reveal key limitations and opportunities in current approaches to grounded visual reasoning.

## 1 Introduction

Recent advances in large vision-language models (LVLMs) have substantially improved visual question answering and reasoning. Proprietary models such as GPT-4V (OpenAI, 2023), Gemini (Google, 2023), and Claude-3 (Anthropic, 2024), along with open-source counterparts like LLaVA (Liu et al., 2023), Qwen-VL (Bai et al., 2023), and DeepSeek-VL (Lu et al., 2024), exhibit strong capabilities in perception, object recognition, and complex visual understanding. However, these successes raise a deeper question: *Do current multimodal models genuinely reason over visual content, or do they rely on superficial patterns and dataset biases?*

Addressing this question requires a closer examination of grounded multimodal cognition—the ability to selectively attend to relevant visual inputs and integrate them into coherent, multi-step logical reasoning. This ability is central to human intelligence and is critical for building interpretable, trustworthy AI systems. For instance, as illustrated in our example in Figure 1, robust reasoning requires the model to justify its answer by explicitly referencing the correct visual regions and following a logical inference path.

Yet, most existing benchmarks primarily focus on end-task performance, offering limited insight into whether models truly "understand" the image or simply exploit dataset biases. They rarely examine how models arrive at answers, whether intermediate reasoning steps align with relevant visual regions (Li et al., 2023a; Liu et al., 2024; Yue et al., 2024; Yu et al., 2024) Recently, increasing attention has been given to the reasoning processes of VLMs, with several works examining their chain-of-thought reasoning (Zhang et al., 2024a; Zhao et al., 2025; Chen et al., 2023), intermediate rationale generation (Chen et al., 2023), and visual grounding consistency (Yang et al., 2023; He et al., 2024). However, these efforts remain fragmented, often lacking unified evaluation protocols and comprehensive grounding diagnostics.

In this work, we introduce a comprehensive benchmark, **MagiC** (Bench**MA**rkin**G** Mult**I**modal **C**ognition), designed to evaluate grounded multimodal cognition. As illustrated in the right section of

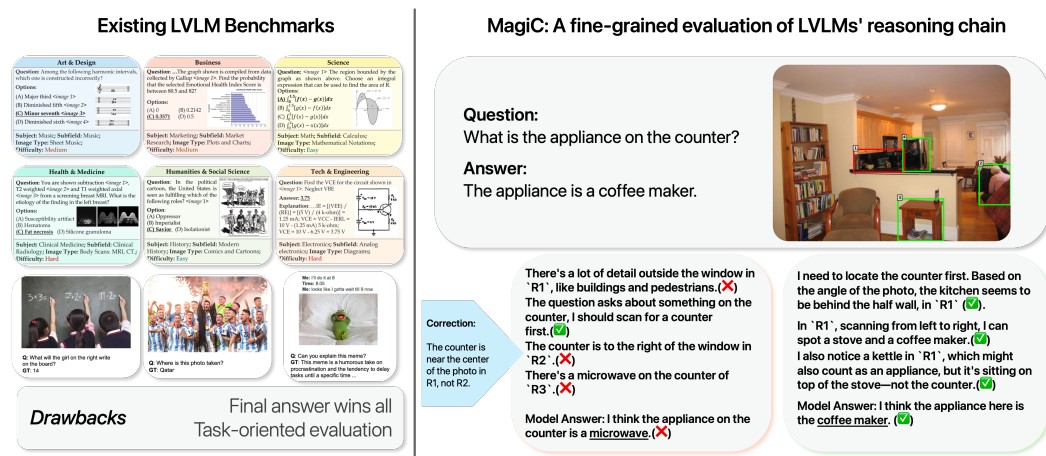

Figure 1: Example scenarios from **MAGIC**. Existing work mainly focuses on evaluating the final answer for a given task ignoring the steps model takes to answer.

Figure 1, our framework goes beyond conventional question answering by measuring a model's ability to (1) generate correct final answers, (2) articulate coherent, interpretable step-by-step reasoning, (3) ground each reasoning step in the appropriate visual evidence, and (4) intervene upon and revise faulty reasoning when possible.

To support this evaluation, we construct a dual-sourced dataset consisting of 5,534 weakly-supervised visual reasoning instances and 896 human-curated instances with over 15,000 annotated reasoning steps. The weakly supervised data is automatically derived from the outputs of high-performing vision-language models across 78 diverse question types, enabling broad coverage with minimal annotation cost. In contrast, the human-curated set includes fine-grained annotations—answers, reasoning rationales, and bounding box groundings—with a held-out test set of 698 instances for rigorous evaluation.

We evaluate 15 vision-language models ranging from 7B to 70B+ parameters, including both open-source and proprietary systems. Our evaluation framework captures four dimensions of model performance: short-form and long-form answer correctness, reasoning validity, grounding quality, and self-correction ability. Grounding quality is quantified using MAGISCORE, which measures the overlap between predicted and reference bounding boxes. To assess models' introspective capabilities, we further analyze their ability to detect and revise intermediate reasoning errors.

In addition to standard performance metrics, **MAGIC** introduces several diagnostic settings that probe robustness and interpretability. The adversarial grounding setting evaluates models under misleading or irrelevant visual cues to test their reliance on correct evidence. The self-correction analysis examines whether a model can recognize and revise its own reasoning errors. Finally, we conduct a human evaluation of reasoning quality on a subset of four models, deriving the *StepSense* score that captures coherence and factual consistency.

In summary, our contributions are as follows:

- First, we introduce a novel task that jointly evaluates answer correctness, step-by-step reasoning quality, and visual grounding fidelity.

- Second, we construct a benchmark dataset comprising 5,534 weakly supervised interleaved bounding-box QA examples and 896 high-quality human-curated examples with detailed annotations for answers, rationales, and bounding boxes.

- Third, we introduce novel evaluation metrics, MAGISCORE tailored to multimodal cognition, with *StepSense* and *Self-Heal* which measures model reasoning quality and self-correction ability.

- Lastly, we conduct a comprehensive analysis of 15 state-of-the-art models, uncovering key insights into both the limitations and potential of current approaches to grounded visual

reasoning. Our findings indicate that models exhibiting precise region focus are generally more likely to answer questions correctly.

We hope this benchmark will catalyze future research toward building more interpretable, robust, and cognitively aligned multimodal systems.

## 2 RELATED WORK

### 2.1 BENCHMARKS FOR LARGE VISION-LANGUAGE MODELS

Ever since the introduction of large vision-language models (LVLMs) such as BLIP (Li et al., 2023b) and LLaVA (Liu et al., 2023), the field has seen rapid advancements in both model scaling and evaluation methodologies. Researchers have increasingly focused on scaling models at test time to push the boundaries of what these systems can achieve. Alongside these developments, there has been a notable surge in the creation of new benchmarks specifically designed to assess the visual reasoning capabilities of LVLMs, moving beyond traditional vision tasks like Visual Question Answering (VQA). Recent benchmarks such as MMBench (Liu et al., 2024), MMMU (Yue et al., 2024), SEED-Bench (Li et al., 2023a), and MM-Vet (Yu et al., 2024) now provide more comprehensive and fine-grained evaluations, covering a wider variety of visual reasoning challenges. However, most of these benchmarks still primarily measure end task performances and do not delve deeply enough into the quality of LVLMs' reasoning process. Despite efforts on visual chain of thought (CoT) (Chen et al., 2023; Rose et al., 2024; Zhang et al., 2024a;b; Chen et al., 2024; Menon et al., 2024; Shao et al., 2024; Zhao et al., 2025; Wu et al., 2025; Xu et al., 2025; Thawakar et al., 2025), evaluating their explicit reasoning quality remains an extremely challenging problem and the current methods leave substantial room for improvement.

### 2.2 GROUNDING IN VISION TASKS

In order to address the gap in existing benchmarks, we examine in particular how well textual reasoning steps remain *grounded in visual input*. Visual grounding has typically been studied in relation to the internal activations of vision encoders and/or in the context of specific visual task (Khan et al., 2022; Yang et al., 2023; He et al., 2024; Reich & Schultz, 2024; Cheng et al., 2024; Wan et al., 2024; Chen et al., 2025) However, our work examines visual grounding more generally in the context of LVLMs' explicit reasoning - if the textual reasoning loses its connection to visual reality, the generated explanations and answers, no matter how fluent, may be incorrect or irrelevant. Our work aims to provide methods for tracing and evaluating this connection throughout the reasoning process.

## 3 THE MAGIC BENCHMARK

We introduce MAGIC, the first benchmark dataset specifically designed to evaluate a model's ability to selectively attend to relevant visual information and integrate it into coherent, multi-step logical reasoning. MAGIC contains **698** test question-and-reasoning pairs, each densely annotated with step-wise correctness labels and corresponding natural-language corrections. The test set encompasses **41** distinct question subtypes derived from the GQA dataset (Hudson & Manning, 2019), categorized into five primary question types: 1) **Verify**: Binary yes/no questions. 2) **Query**: Open-ended questions. 3) **Choose**: Questions offering a choice between two alternatives (e.g., "Is it red or blue?"). 4) **Logical**: Questions requiring logical inference. 5) **Compare**: Questions involving comparisons between two or more objects.

Detailed statistics of the dataset are summarized in Table 1. In total, we collected 896 annotated questions, comprising correctness annotations for 8403 individual reasoning steps and 2700 natural-language corrections. In which, 198 are reserved for development and 698 for test, and the test set is used as the MAGIC benchmark. The distribution of the question types of the test set can be found at Figure 2.

The construction of MAGIC involves three main steps: task-input processing, response collection, and human correction collection. Examples of the dataset, are available in the Appendix B.

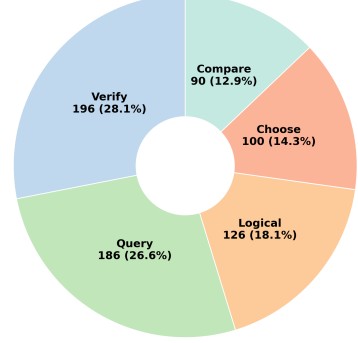

| Set | WS* | Dev | Test |
|---|---|---|---|
| **# of Instances** | 5534 | 198 | 698 |
| **# of Steps** | 69523 | 2373 | 8403 |
| Avg. Words/Step | 23 | 18 | 19 |
| Avg. Steps/Inst. | 12.00 | 11.98 | 12.03 |
| **# of Correction** | N/A | 641 | 2700 |
| Avg. Words | N/A | 18 | 19 |

Table 1: Dataset statistics.
*WS: Weakly supervised split*

Figure 2: Question type distributions. *Detailed distribution of detailed types can be found in Figure 6.*

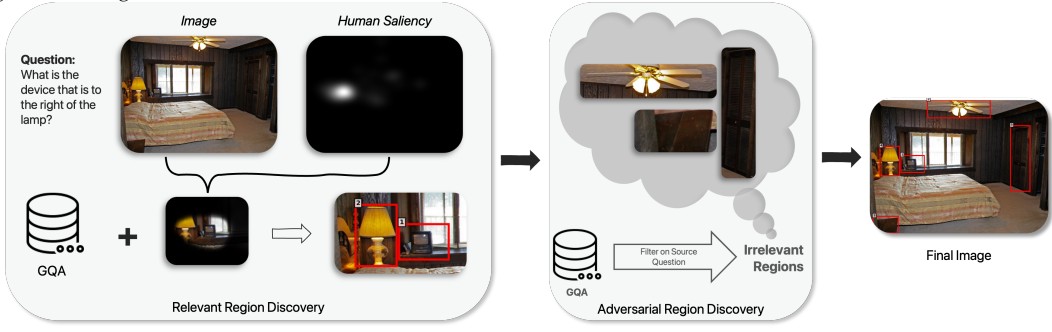

Figure 3: Illustration of Task-Input Data Construction Processing. *Human saliency maps are used only during construction to identify relevant regions and are never provided to LVLMs at evaluation time.*

## 3.1 TASK-INPUT CONSTRUCTION

We obtain images ($I$), questions ($Q$), and short/full ground-truth answers ($GT_{short}$, $GT_{full}$) from the GQA dataset (Hudson & Manning, 2019). We restrict our selection to the public `val` splits aim to minimize the likelihood of dataset overlap with existing model knowledge. Additionally, we incorporate human saliency maps ($Saliency_q$) from AiR-D (Chen et al., 2020), which are based on human eye-tracking data, providing higher quality saliency regions compared to pointer-based counterparts. Although AiR-D contains saliency maps corresponding to both correct and incorrect human answers, we only utilize those maps associated with correct responses.

As illustrated in the relevant region discovery section in Figure 3, for each question $Q$ and its associated saliency map $Saliency_q$, we identify candidate "hotspot" regions that received significant human attention. Bounding boxes for these regions are extracted and merged with the annotated objects from GQA, forming an initial set of relevant regions. These regions are further processed to eliminate overlaps or merge boxes where necessary, resulting in the final set of relevant bounding boxes, $RBox_q$, for question $Q$.

Next, building upon the identified relevant regions ($RBox_q$) and GQA's image-level object annotations, we select additional image regions relevant to other GQA questions but irrelevant to the current question $Q$, as demonstrated in Figure 3. These are termed adversarial regions. We randomly sample adversarial regions that do not overlap with any region in $RBox_q$, creating the set $ABox_q$. To ensure sufficient adversarial content, we sample three adversarial regions per question; if fewer than three such regions exist, additional regions are generated randomly. We henceforth refer to $Box_q = RBox_q \cup ABox_q$ as the candidate box set and to the labels indicating which boxes are relevant as the ground-truth relevance. A detailed implementation can be found in algorithm 1.

This step yields tuples of the form $(I_q, Box_q, Q_q, A_q)$ for each question $q$.

## 3.2 LVLM RESPONSE COLLECTION

Using the processed input data, we collect responses from four LVLMs representing four different model families: INTERNVL-2.5 8B, GEMMA 3 27B, QWEN2.5-VL 7B, and OPENAI GPT-4O.

Each model is prompted using in-context learning (ICL) examples created in-house, provided with inputs $(I_q, Q_q)$, and tasked to generate step-by-step reasoning from the first-person perspective, concluding with a final answer.

Responses exhibiting invalid formatting or repetitive reasoning loops are filtered out. Remaining valid responses are segmented into individual sentences, each representing a distinct reasoning step $S_i$. From these valid responses, we randomly select one reasoning sequence per question $(S_q)$, which is then densely annotated by human annotators.

### 3.3 REASONING CORRECTION COLLECTION

We recruit four expert annotators (computer science students with relevant experience) to evaluate and correct the sampled model responses. Annotators begin by annotating ten practice examples, which serve as training materials and are discarded thereafter. In the annotation phase, annotators first verify the quality of each question and compare the model's final answer against the ground truth. Subsequently, they assess the correctness of each individual reasoning step $S_i$. This annotation archives a substantial inter-annotator agreement of 0.72 Cohen's Kappa. Annotators are required to document any ambiguities, poorly formulated questions, or reasoning steps requiring reconsideration. A discussion session is conducted after every 100 annotated examples to resolve disagreements and collectively refine annotations.

For each incorrect reasoning step $S_i$, annotators will also provide a corrected natural-language version. These corrections are complete rewrites rather than simple critiques, intended to maintain coherent and logical reasoning when replacing the original step.

After completing this annotation step, each question $Q$ is associated with an annotated reasoning sequence $S_q = \{s_1, s_2, \ldots, s_n\}$, where each element $s_n$ consists of the original reasoning step, a correctness label, and its corresponding correction $s_n = (original, correctness, correction)$.

### 3.4 WEAKLY SUPERVISED DATASET

In addition to the human-annotated dataset, we also created a weakly supervised dataset following the pipeline similar to that described in Section 3.1. The weakly-supervised split omits human saliency but derived its boxes from GQA's object bounding boxes. To avoid overlap with our test set, images included in this dataset are distinct from those in the primary annotated set. Model responses in this version were collected using inputs $(I_q, Q_q, A_q)$ without adversarial regions to produce cleaner reasoning data.

## 4 TASK OVERVIEW

Using an image-question pair $(I, Q)$ from the **MAGIC** benchmark, we sample model responses from Large Vision-Language Models (LVLMs), denoted by $S_{\mathcal{LM}}$. The response from a given model can be represented as $S_{\mathcal{LM}} = [s_1, s_2, \ldots, s_n]$, where $n$ is the number of reasoning steps generated by the model, along with the model's final answer $A_{\mathcal{LM}}$. Additionally, the dataset provides ground-truth bounding boxes $Box_q$ and corresponding ground-truth answers in both short and long form, denoted by $(GT_{short}, GT_{full})$. Each question $Q$ also includes a human-corrected reasoning chain, $S_q$. Utilizing the tuple $(I, Q, Box_q, S_{\mathcal{LM}}, A_{\mathcal{LM}}, GT_{short}, GT_{full}, S_q)$, we evaluate LVLMs on the following three tasks:

### 4.1 REGION FOCUS

For each question $Q$, a LVLM generates a reasoning chain $S_{\mathcal{LM}} = [s_1, s_2, \ldots, s_n]$. We introduce the region extractor function $\phi(\cdot)$ that maps each reasoning step $s_i$ to a set of bounding box indices from $Box_q$, defined as $\phi(s_i) \subseteq \{1, \ldots, |Box_q|\}$. By taking the union of these regions across all reasoning steps, we derive the complete set of regions the model focused on as part of its reasoning: $\hat{\mathcal{R}} = \bigcup_{i=1}^{n} \phi(s_i)$.

We can then represent the set of unique regions explored by the model as a binary vector: $\hat{\mathbf{y}} \in \{0, 1\}^{|Box_q|}$, where $\hat{\mathbf{y}}_k = \mathbb{1}[k \in \hat{\mathcal{R}}]$. Here, $\hat{\mathbf{y}}_k = 1$ indicates that at least one reasoning step

utilizing the $k$-th bounding box. Analogously, we have the ground-truth vector from the benchmark: $\mathbf{y}^* \in \{0,1\}^{|Box_q|}$, where $\quad y_k^* = 1$ if and only if the $k$-th box in $Box_q$ is labeled as relevant.

Using $\hat{\mathbf{y}}$ and $\mathbf{y}^*$, we evaluate two variants of region-focus scores:

MAGISCORE (micro): Aggregating predictions across all images, we calculate precision, recall, and $F_1$ scores based on cumulative totals of true positives, false positives, and false negatives. MAGISCORE (macro): For each question $Q$, we first compute image-level precision, recall, and $F_1$ individually. The macro-level score is then obtained by averaging these metrics across all questions.

Note that the primary goal of MAGISCORE is to assess visual grounding and reasoning capabilities independently of object detection performance, making the metrics distinct from the detection-oriented measures such as Intersection over Union (IoU). Detailed illustration of the score calculation is in Appendix D.

## 4.2 LLM-ASSISTED FINAL ANSWER ACCURACY

While Hudson & Manning (2019) propose exact match as the evaluation metric for answer accuracy, this approach has notable limitations when evaluating natural language responses generated by LVLMs. Specifically, exact match fails to recognize synonymous or semantically equivalent responses (e.g., "tan" vs. "brown") and may incorrectly reward random occurrences of the ground-truth term within longer responses. Therefore, we propose an LLM-assisted evaluation approach by leveraging a large language model as a judge to more effectively assess correctness.

The LLM-based judge is modeled as the following: $f_{\mathcal{LM}} : P(A_{\mathcal{LM}}, GT_{full}) \mapsto \{0,1\}$

This function classifies a model's predicted answer $A_{\mathcal{LM}}$ against a provided ground-truth answer. Using this function, we define two accuracy measures:

**1. Short Answer Accuracy** (**Acc$_{\text{short}}$**): evaluates whether the predicted answer captures essential elements indicated by the short ground-truth answer: $\text{Acc}_{\text{short}} = \frac{1}{|\mathcal{D}|} \sum_{(I,Q) \in \mathcal{D}} f_{\mathcal{LM}}\big(A_{\mathcal{LM}}, GT_{\text{short}}\big)$

**2. Full Answer Accuracy** (**Acc$_{\text{full}}$**): evaluates whether the predicted answer aligns comprehensively with the detailed ground-truth answer: $\text{Acc}_{\text{full}} = \frac{1}{|\mathcal{D}|} \sum_{(I,Q) \in \mathcal{D}} f_{\mathcal{LM}}\big(A_{\mathcal{LM}}, GT_{\text{full}}\big)$

The short-answer accuracy measure can be viewed as a more permissive evaluation metric compared to full-answer accuracy, as it only requires capturing core semantic elements identified by the concise ground-truth answer.

## 4.3 SELF-CORRECTION

For each question $Q$, the benchmark provides human-corrected reasoning chains, denoted as $S_q = [s_1, s_2, \ldots, s_n]$, where each step has been labeled with correctness and natural language correction if is incorrect. Given the annotated reasoning chain $S_q$, we identify contiguous subsequences of incorrect reasoning steps marked by human annotators as $S_{wrong} = [s_i, \ldots, s_j]$, with indices $2 \leq i \leq j \leq n-2$.

To evaluate the self-correction ability of the LVLM, we design an intervention experiment. Specifically, we inject the initial reasoning steps up to and including the incorrect step, i.e., the subsequence $[s_1, s_2, \ldots, s_j]$, into model's response. With this partial reasoning chain as context, the LVLM continues to generate subsequent reasoning steps, resulting in an extended reasoning chain defined as $S_{rest} = [s'_{j+1}, s'_{j+2}, \ldots, s'_{n'}]$. Here, $S_{rest}$ represents the model's attempt to self-correct previously identified erroneous reasoning while continuing to answer the given question $Q$.

We again employ an LLM-based judge to identify the presence of these human corrections within the model-generated steps in $S_{rest}$. The model returns 1 if any of the reasoning steps within $S_{rest}$ semantically match the corresponding human correction steps from $[s_i, s_{i+1}, \ldots, s_j]$, and 0 otherwise.

Aggregating across all the incorrect subsequences identified by annotators, we compute the score for *Self-Heal* as the fraction of instances in which the LVLM successfully corrects the injected erroneous reasoning step.

# 5 EXPERIMENT AND RESULTS

In this section, we perform a series of experiments on multiple state-of-the-art LVLMs to examine their visual grounding abilities and overall reasoning quality. We found that accuracy climbs almost linearly with how model's focus aligns to relevant regions, and scaling boosts its performance on region focusing, reasoning and self-correction.

## 5.1 EXPERIMENTAL SETTINGS

We evaluate a total of 12 open-source, general-purpose LVLMs without test-time inference scaling capabilities, including LLaVA-OneVision-Chat (Xiong et al., 2024), InternVL 2.5-MPO (Wang et al., 2024), Qwen2.5-VL (Bai et al., 2025), Llama-3.2 (AI@Meta, 2024), Aya Vision (CohereForAI, 2025), and Gemma 3(Team, 2025). Additionally, we evaluate the only open-source model equipped with test-time inference scaling to the best of our knowledge, QvQ (QwenTeam, 2024)[1], and two proprietary LVLMs from OpenAI: GPT-4o-mini (Hurst et al., 2024) and GPT-4.1 (OpenAI, 2025).

As described in section 4, for each image-question pair $(I, Q)$ in the test set, models are instructed to generate detailed reasoning steps required to answer the provided question, based on the associated image with clearly marked bounding boxes. As constructed in section 3.1, the test set includes both relevant and adversarial boxes by default, and models are evaluated under this mixed candidate set. Both the reasoning steps and final answers produced by each model are extracted for evaluation. Additional experimental details can be found in the Appendix C.

| Model | Size | Think Capability | Final Answer Full | Short | MAGISCORE (Macro) Precision | Recall | F1 | MAGISCORE (Micro) Precision | Recall | F1 |
|---|---|---|---|---|---|---|---|---|---|---|
| *Small (∼7B) Open LVLM* | | | | | | | | | | |
| LLaVA OneVision | 7B | No | 49.00 | 54.01 | 48.31 | 35.33 | 37.79 | 77.78 | 32.72 | 46.06 |
| InternVL 2.5-MPO | 8B | No | 49.86 | 50.00 | 65.61 | 53.50 | 54.12 | 66.61 | 50.90 | 57.71 |
| Aya Vision | 8B | No | 45.13 | 50.00 | 78.63 | 50.32 | 56.74 | 89.02 | 43.83 | 58.74 |
| InternVL 3 | 9B | No | 49.43 | **56.45** | 69.62 | 58.09 | **58.31** | 66.60 | 53.34 | **59.24** |
| Qwen2.5-VL | 7B | No | **51.00** | 55.30 | 53.49 | 42.60 | 43.15 | 67.93 | 38.51 | 49.15 |
| *Medium (11∼32B) Open LVLM* | | | | | | | | | | |
| Gemma 3 | 12B | No | 50.00 | 54.58 | 76.49 | 69.15 | **67.69** | 73.03 | 64.45 | **68.47** |
| InternVL 2.5-MPO | 26B | No | 36.39 | 40.69 | 71.47 | 58.17 | 59.44 | 76.59 | 53.30 | 62.86 |
| Gemma 3 | 27B | No | 50.29 | 52.44 | 69.46 | 73.23 | 66.48 | 66.61 | 70.07 | 68.30 |
| Aya Vision | 32B | No | 47.71 | 52.44 | 84.68 | 57.23 | 63.15 | 87.74 | 51.24 | 64.70 |
| Qwen2.5-VL | 32B | No | **59.74** | **63.32** | 73.85 | 57.03 | 59.18 | 69.07 | 51.42 | 58.95 |
| *Large (72∼90B) Open LVLM* | | | | | | | | | | |
| Llama 3.2 Vision | 90B | No | 35.67 | 39.97 | 46.28 | 37.55 | 38.29 | 83.35 | 33.49 | 47.78 |
| Qwen2.5-VL | 72B | No | **61.03** | **66.05** | 60.80 | 80.95 | **63.37** | 56.82 | 75.43 | 64.81 |
| QvQ | 72B | Yes | 59.89 | 64.04 | 47.43 | 89.35 | 60.26 | 53.51 | 88.46 | **66.69** |
| *Proprietary LVLMs* | | | | | | | | | | |
| GPT 4o mini | N/A | No | 51.72 | 55.01 | 69.03 | 79.14 | **66.77** | 63.88 | 67.71 | 65.74 |
| GPT 4.1 | N/A | No | **71.78** | **74.36** | 65.98 | 71.10 | 63.73 | 61.62 | 74.01 | **67.25** |

Table 2: Performance comparison. The best performing models for each metrics are shown in **bold**.

## 5.2 MAIN RESULTS

**Focusing on the right place leads to higher answer performance.** Across the 15 LVLMs evaluated in Table 2, we find that the models with higher MAGISCORE tend to perform better in answering the question itself. This pattern holds for both small- and large-scales, suggesting that the ability to selectively attend to relevant regions is a strong indicator for the correctness of models' final answer. For example, GEMMA 3 12B, which scored 67.7% in attention, achieved 54.6% in short answer accuracy, while LLAMA 3.2-90B, scored 38.3% in attention only and subsequently achieved 39.97% in answer accuracy. We notice that the QWEN2.5-VL models generally tend to have lower attention scores than their peers despite high short-answer accuracies. We speculate that the attention scores gap with other models probably comes from Qwen's training recipe (Bai et al., 2025), which includes pre-training on various VQA datasets, and grounding data with absolute position references, which

---

[1]Note that we do not provide QvQ with ICL examples unlike other models, as by default QvQ explicitly generates its intermediate reasoning towards the answer.

may interfere with the models' ability to make *relative* references using specific schemes such as our bounding boxes.

**Region focus as an emergent ability of larger models.** As shown in Figure 4, we observe that scaling model sizes could lead to better performance in utilizing the relevant regions. Medium-sized (11-32B) LVLMs are on the sweet spot in terms of both region attention performance and final answer performance, as we can see that they generally tend to have higher region attention scores than smaller models. This is followed by higher final answer scores, in terms of both full and short answers. Large-sized (72B+) models seem to provide further performance gains over medium-sized models. For QWEN2.5-VL models (7B, 32B, 72B) with lower attention scores than their respective peers, we can still see that the overall trend still holds within these three models.

**Attention capabilities of test-time scaling models.** It is interesting to note that the only vision test-time scaling model we test, QVQ, achieves comparable performance to QWEN2.5-VL 72B, despite not being provided with our in-context learning examples of utilizing bounding boxes for reasoning. We can see that as part of its test-time scaling, QVQ could already generate reasoning that effectively considers the bounding boxes we inject into test images. However, the model tends to perform exhaustive coverage of all bounding boxes, as evidenced by low precision in micro and macro attention scores. While it remains to be seen whether this would be the case for other comparable models, we see a great potential of better visual grounding to be achieved with test-time scaling.

| Model | Size | Accuracy | |
|---|---|---|---|
| | | *StepSense* | Answer |
| QwenVL 2.5 | 7B | 61.38 | 69.98 |
| InternVL 2.5 | 8B | 48.99 | 48.69 |
| Gemma 3 | 27B | 52.13 | 55.1 |
| GPT 4o mini | N/A | 53.89 | 59.26 |

Table 3: *StepSense* and final answer accuracy

| Model | Size | *Self-Heal* |
|---|---|---|
| LLaVA OneVision | 7B | 25.99 |
| Gemma 3 | 12B | 31.5 |
| Qwen2.5-VL | 7B | 35.03 |
| Qwen2.5-VL | 32B | 41.38 |
| Qwen2.5-VL | 72B | 46.89 |

Table 4: *Self-Heal* accuracy

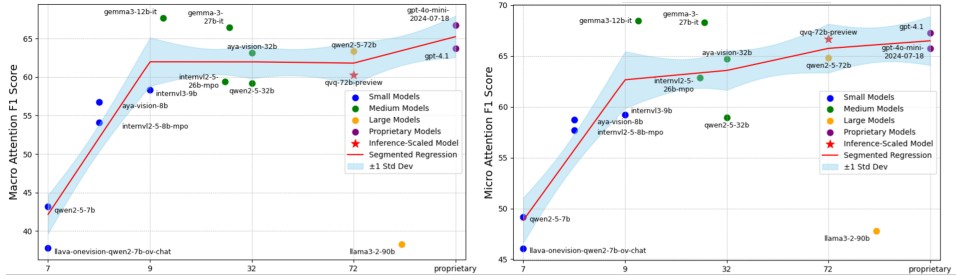

Figure 4: Scaling curve for MAGISCORE

**Sound intermediate reasoning, sound final answer.** Human-annotated evaluation from Table 3 shows that models with higher sentence-level reasoning accuracy almost always enjoy higher final-answer accuracy - for example, QWEN2.5-VL 7B leads both metrics (61.4% vs. 70.0%), while INTERNVL 8B trails on both. Together with our findings in region focus, this supports a three-step chain: a precise perception with reliable reasoning can result in correct answers. Improving any links in that chain, and especially the visual grounding link, therefore remains a direct path to better overall performance.

**Stronger models correct themselves better.** The self-correction results in Table 4 echo the same scaling story. Each jump in QWEN2.5-VL size yields roughly a six-point boost in correction accuracy. We also tested two models, LLaVA-OneVision 7B and Gemma 3 12B, which achieved short answer performance similar to QWEN2.5-VL 7B. However, we can see that both models lag noticeably behind QWEN2.5-VL 7B, showing that stronger model's tend to have better self-correction capabilities. as well.

### 5.3 ERROR ANALYSIS

We randomly sample 100 instances of failed reasoning from all 15 LVLMs we have tested and analyze their failure points. Figure 5 shows a list of failed examples. We identify the following common error patterns:

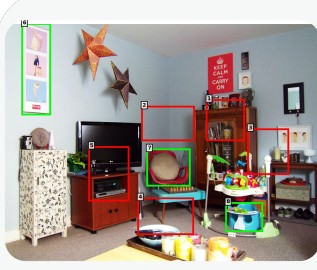 I'm looking for furniture, so I should look at the base of this TV. **There's a large, dark wooden piece of furniture underneath it, in `R2`.** It appears to be a cabinet or a console of some sort.

Incorrect object location (Gemma 3)

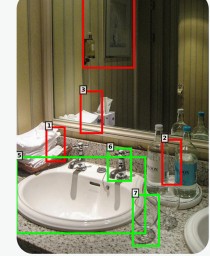 Let me start by checking the bounding boxes. In `R4`, I see a dispenser mounted on the wall. It looks to be a soap or lotion dispenser. **Next, I should search for any painting nearby.**

Incorrect region description (GPT 4o-mini)

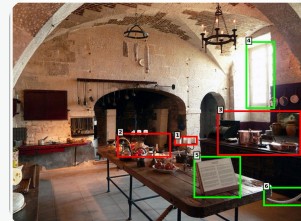 Next, I'll look at region `R2`, **which is to the right of region `R1`.** This region may be part of a shelf or another segment of the tabletop.

Incorrect spatial orientation (InternVL2.5)

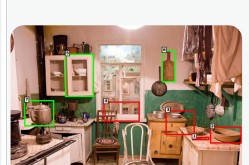 Looking at the bounding boxes:

1. Box 1 is around a wooden bowl on a countertop to the right.
2. Box 2 is near some hanging utensils.
3. Box 3 is around a wooden cabinet or dresser.
4. Box 4 is near a chair close to the window.
5. Box 5 is on a wall cabinet to the left.
6. Box 6 is around a hanging spatula or similar utensil.
7. Box 7 is around a samovar or tea pot on the left side.

The question mentions a wood bowl and a utensil on the right. So, I need to identify the positions of these specific items.

Exhaustive use of bounding boxes (QVQ)

Figure 5: **Examples of failed reasoning.** Adversarial boxes are labeled in green.

**Exhaustive coverage of all regions** In many cases, the models tend to exhaustively consider all bounding boxes that appear in the image, despite our instruction to selectively utilize bounding boxes that are relevant for answering the question. Moreover, it is often found that models decide to continue iterating through every single bounding boxes, even though it has already inspected other bounding boxes and realized the answer to the question. This is a potentially problematic behavior as it is not only inefficient to consider irrelevant regions, but also could increase the chances of hallucination over analysis into regions that are actually relevant. We find that this is especially the case for QvQ, which is trained to generate longer chains of reasoning by default.

**Incorrect object location** Another pattern we have identified is that even though the models correctly identified relevant details required to answer the question, they often hallucinate by claiming that such details appear in unrelated bounding boxes. For example, in the top left example of Figure 5, it is true that there is a dark wood piece of furniture under the TV. However, the model incorrectly claims that it is located in R2, not R5.

**Wrong spatial relations between the bounding boxes** Related to the issue of incorrect object location, we also see that the models often seem to experience issues with spatial orientation, as evidenced by the bottom left image of Figure 5. While R2 is actually located to the left of region R1, the model incorrectly claims that it is actually to the *right* of R1.

**Incorrect region description** Even when the model has identified and analyzed relevant bounding boxes, we found that the model output incorrect details of the region. In the bottom right image of Figure 5, the model was able to correctly choose R4 as part of its analysis and identifies the soap dispenser in the region. However, it fails to see that the painting in question is also in R4.

## 6 CONCLUSION

This paper introduced **MAGIC**, a novel benchmark designed to evaluate grounded multimodal cognition. We evaluated 15 LVLMs ranging from 7B to 70B+ parameters across four dimensions: final answer correctness, reasoning validity, grounding fidelity, and self-correction ability. Our analysis shows that the models that could focus better on relevant regions tend to achieve better task performance, and such focusing ability improves with increased model sizes. However, we also found that current SOTA LVLMs all suffer from various types of region focus failures, including incorrect object location, wrong spatial relations, and inaccurate region descriptions.

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

## A  DATA STATISTICS

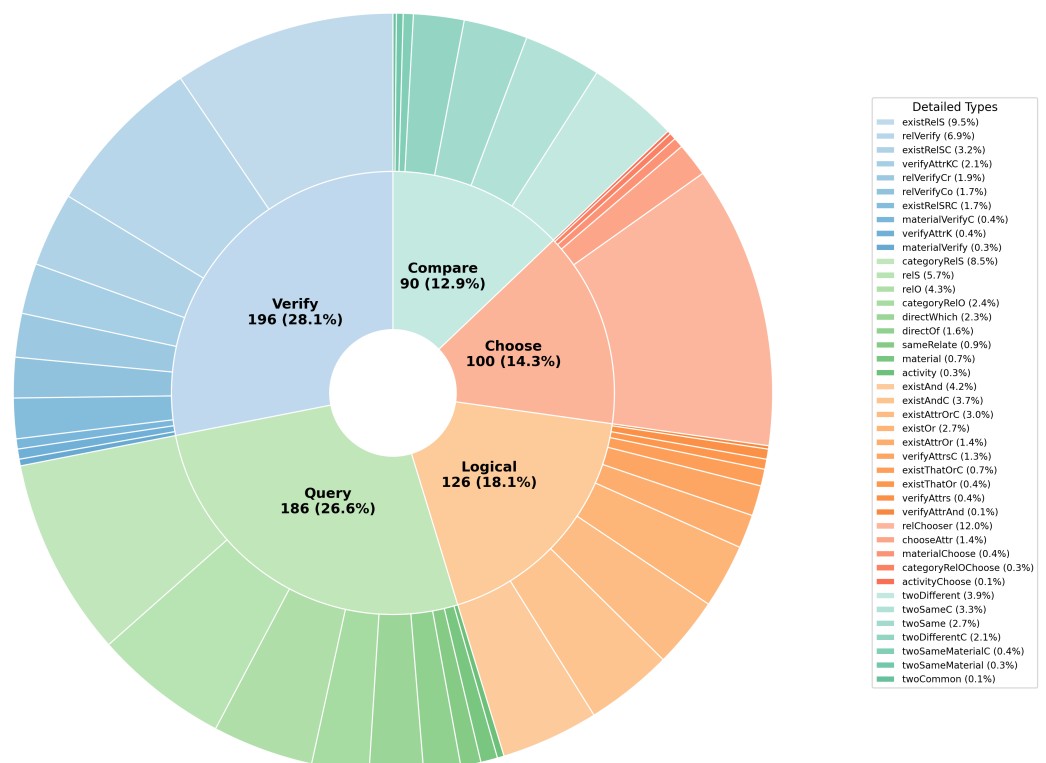

Figure 6: Detailed question type distributions for **MAGIC** benchmark.

## B  DATA EXAMPLES

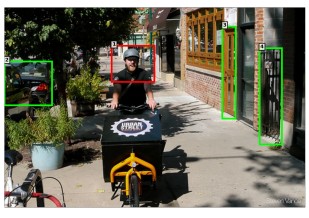

**Question:** Is there a snowboard or a helmet in the picture?

**Ground Truth:** Yes, there is a helmet.

**Model Answer:** Yes, there is a helmet. ✅

Legend: [Correct] [Wrong]

**Reasoning Steps:**

**1.** Okay, so the question asks if there's a snowboard or a helmet in the picture.

**2.** Let's start by looking at the whole image to get a sense of the scene.

**3.** It looks like an outdoor setting with a person on a bicycle and some buildings nearby.

**4.** I should start by scanning the image for either a snowboard or a helmet.

**5.** Since bicycles are usually associated with helmets, I'll start by looking at the person in the image.

**6.** I see a person riding a bicycle in the center of the image, near *Region 1*.

**7.** Let me take a closer look at them.

**8.** Zooming in on *Region 1*, I can see the person is wearing a helmet.

**9.** That answers the question—there is definitely a helmet in the picture!

**10.** I don't see any snowboards anywhere in the image, though.

**11.** So, no need to look further.

**12.** I can confirm that the question is answered with just the helmet, since there isn't a snowboard in sight.

Figure 7: Example where both reasoning and answer are correct

**Reasoning Steps:**

**1.** Okay, I need to figure out what is in the pot to the right of the person.

**2.** Let me first take a look at the image to see what we've got.

**3.** It looks like there's a body of water, a bench, and people sitting around tables under umbrellas.

**4.** Now, I need to find a person and then look to their right for a pot.

**5.** I start by scanning for people.

**6.** There are several people gathered around region *Region 2*, which seems to be the main activity area.

**7.** They are seated and standing around tables with umbrellas.

**8.** I focus on the main activity area and notice a person sitting close to region *Region 2*.

**9.** To their right, there is a table, and on top of that table, there is a pot. **Correction**: To their right, there is a big pot on the ground in *Region 1*.

**10.** I see some greenery in the pot that appears to be flowers.

**11.** So, there is a pot on the table to the right of the person, and it contains flowers. **Correction**: So, there is a pot on the ground to the right of the person, and it contains flowers.

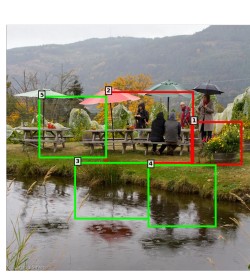

**Question:** What is in the pot to the right of the person?

**Ground Truth:** The flowers are in the pot.

**Model Answer:** The pot contains flowers. ✅

**Legend:** Correct | Wrong

Figure 8: Example where reasoning is partially incorrect and answer are correct

**Reasoning Steps:**

**1.** Okay, I need to figure out what appliance is on top of the counter.

**2.** Let me take a look at the image.

**3.** It looks like a room with a kitchen area visible.

**4.** There's a person sitting and another one standing, so it seems like it's someone's house.

**5.** Now, let's see where the counter is located.

**6.** It's probably in the middle of the room.

**7.** I can see a white counter in the image, and it's located in the center part of the picture, starting around *Region 1*.

**8.** The question asks about an appliance on top of the counter.

**9.** Scanning *Region 1*, I can spot a microwave. **Correction**: Scanning *Region 2*, I see a white toaster on top of the counter.

**10.** It seems to be placed directly on top of the counter.

**11.** It's a black microwave oven.

**12.** There's also a toaster on the counter to the left of the microwave.

**13.** I should probably focus on the microwave as that seems like the primary appliance.

**14.** I'm pretty confident now.

**15.** The appliance on top of the counter is a microwave. **Correction**: Thus, the appliance is a toaster.

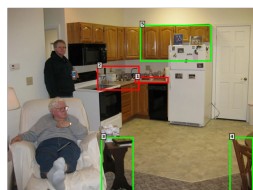

**Question:** What is the appliance on top of the counter?

**Ground Truth:** The appliance is a toaster.

**Model Answer:** The appliance is a **microwave**. ❌

**Legend:** Correct | Wrong

Figure 9: Example where both reasoning and answer are incorrect, red boxes without correction means manual steer of reasoning from annotator.

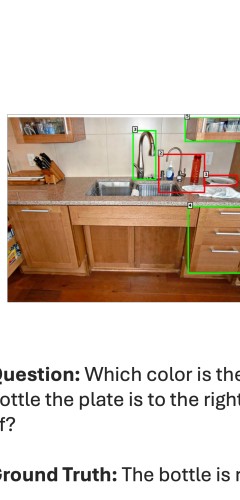

**Reasoning Steps:**

1. Okay, I need to find the color of the bottle that is to the right of the plate.

2. Let me take a look at the image.

3. This appears to be a kitchen scene, with a counter and some appliances.

4. I need to locate the plate and then identify the bottle immediately to its right.

5. Looking around, I see a counter with a lot of items on it in Region 1 and Region 2.

6. So, I'm going to start scanning there.

7. From that perspective, I see a white plate in *Region 2*.
**Correction**: From that perspective, I see a white plate in *Region 1*.

8. To the right of that plate, there is a red bottle.
**Correction**: To the left of that plate, there is a red bottle.

9. I think the color of the bottle to the right of the plate is red.
**Correction**: I think the color of the bottle to the left of the plate is red.

**Question:** Which color is the bottle the plate is to the right of?

**Ground Truth:** The bottle is red.

**Model Answer:** The bottle is red. ✅

Legend:    Correct    Wrong

Figure 10: Example where reasoning is partially incorrect and answer are correct

## C    REPRODUCIBILITY

We evaluate 13 open LVLMs and 2 proprietary LVLMs as shown in Table 2. We use `vllm` on all models except `LLaVA`, `Aya Vision`, and `InternVL` which uses `transformers` library. We set the sampling temperature with the default temperature provided with the each model for the three main tasks. All experiments are performed on either a single Nvidia A100-80GB GPU or Nvidia A100-40GB GPU.

LLM-assisted evaluation uses `Qwen/Qwen2.5-72B-Instruct-AWQ` with greedy decoding (`temp=0`) using `vllm` on a single Nvidia A100-SXM-80GB GPU.

## D    MAGISCORE CALCULATION EXAMPLE

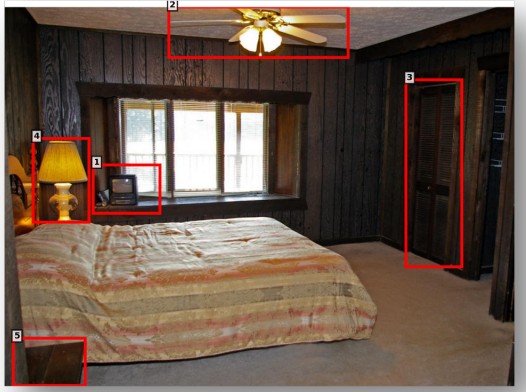

Relevant regions: [1, 4]
$$y^* = [1, 0, 0, 1, 0]$$

Used regions: [1, 2, 3]
$$\hat{y} = [1, 1, 1, 0, 0]$$

$$TP = 1; FP = 2; TN = 1; FN = 1$$
$$Precision = \frac{1}{3}; Recall = \frac{1}{2}; F1 = 0.4$$

Figure 11: Example on how MAGISCORE are calculated.

# E    DETAILS ON TASK-INPUT PROCESSING

---

**Algorithm 1:** Generate and Save Relevant Bounding Boxes

---

**Function** `FindBboxes`(*s_path*, *img_size*)**:**
   $M \leftarrow$ load_gray($s\_path$) $\rightarrow$ resize($img\_size$) $\rightarrow [0,1]$
   $B \leftarrow \big(M >$ thresh$\big)$                               `// binarize`
   $B \leftarrow$ morph_clean($B$)
   **return** all bbox($r$) for each connected region $r$ with area($r$) $\geq area_{min}$
**return**
**Function** `RemoveOverlap`($\mathcal{B}$)**:**
   **return** $\mathcal{B}$ pruned by IoU / coverage / temporal rule at threshold
**return**
**Function** `AdvBoxes`(*id*, $\mathcal{G}$, $S$)**:**
   $\mathcal{C} \leftarrow$ cached_adv($id$)
   $\mathcal{C} \leftarrow$`RemoveOverlap`($\mathcal{C} \cup \mathcal{G}$)
   $\mathcal{C} \leftarrow$ size_filter($\mathcal{C}$)
   **return** sample or synthesize boxes so that $|\mathcal{C}|$ meets the required quota
**return**

**foreach** *question* $(qid, q)$ *in* `questions` **do**
   $I \leftarrow$ read_image($q$.imageId)
   $\mathcal{S} \leftarrow \varnothing$
   **for** $t \in \{0, 1, 2\}$ **do**
      $m \leftarrow$ load_saliency_map($qid$, $t$)
      $\mathcal{S} \leftarrow \mathcal{S} \cup$`FindBboxes`($m$, $I.size$)
   **end**
   `label` $\mathcal{S}$ as *non_adv*
   $\mathcal{G} \leftarrow$ ground–truth objects (gqa objects)
   $\mathcal{B} \leftarrow$ merge($\mathcal{G}$, $\mathcal{S}$)
   $\mathcal{B} \leftarrow$`RemoveOverlap`($\mathcal{B}$, *coverage*, *user_thr*)
   $\mathcal{B} \leftarrow \mathcal{B} \cup$`AdvBoxes`($q.imageId$, $\mathcal{B}$, $I.size$)
   `draw&save`($I$, $\mathcal{B}$)
**end**

---

# F    LIMITATION

A primary limitation of this work is its reliance on the GQA dataset, largely due to the limited availability of publicly accessible eye-tracking data and richly annotated scene graphs. Consequently, the findings may be influenced by certain dataset-specific characteristics and may not encompass the entire spectrum of real-world scenarios, such as answering math questions in images or OCR tasks. While this study offers an important first step toward understanding LVLMs' multimodal cognition abilities, this limitation highlights opportunities for future research.

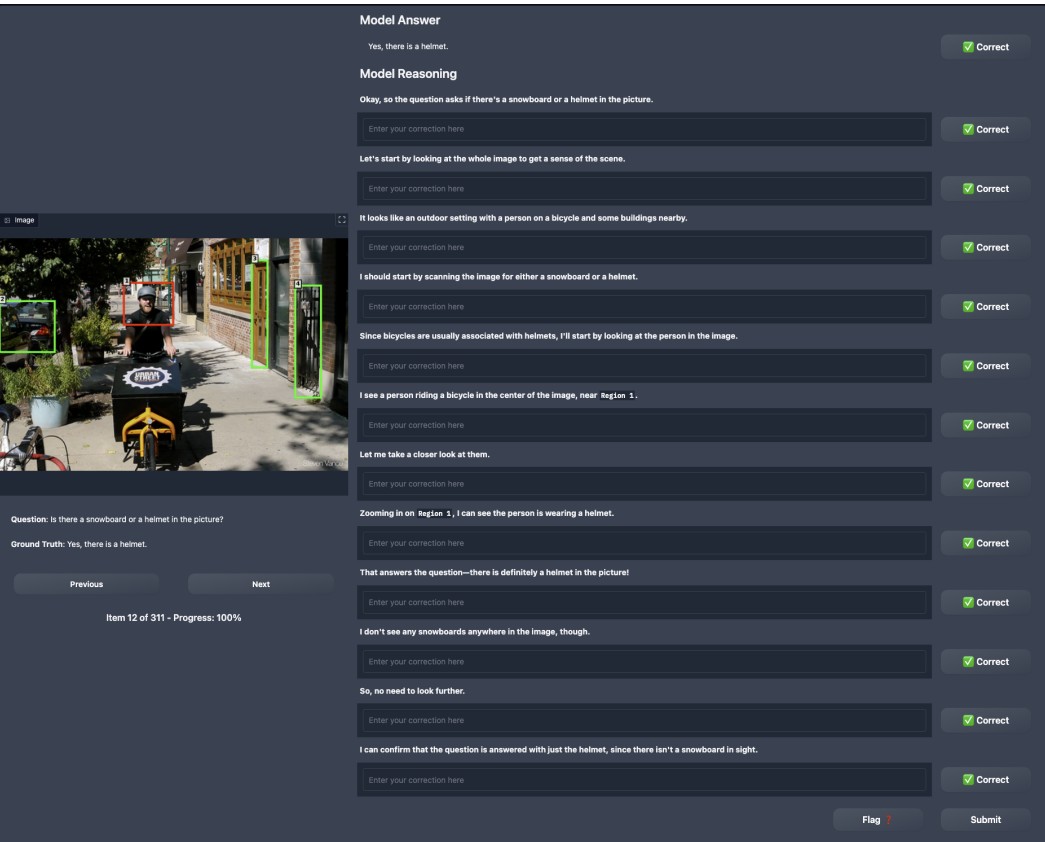

Figure 12: Annotation User Interface

**In-Context Learning examples go here**
Similarly to the examples provided above, please imagine this person's thought process behind the areas of the image the person looked at in order to solve the question, which are marked with the red bounding boxes.
The thought process should meet the following conditions:
- It should describe the broader setting of the image and how that context informs the search for specific details.
- It should explain how the bounding boxes (e.g., 'R1', 'R2') were used to narrow down the areas of interest. For instance, it should mention how the image's orientation or the placement of objects guided the focus to certain regions.
- It should only consider the bounding boxes that does appear in the image and not refer to any bounding boxes that actually do not exist. Note that all bounding boxes are numbered sequentially, so if we have 4 bounding boxes in the image, there are exactly the following bounding boxes: 'R1', 'R2', 'R3', and 'R4'.
- It should list specific visual cues noticed within each highlighted region and how these observations lead to identifying the relevant subject (e.g., a man with a backpack).
- It should clearly connect each observation to the final answer. It should explain how, based on the evidence gathered from the image, the conclusion was reached (e.g., identifying the man holding a cell phone).

Write as if you are this person, in a similar style as above examples.

## Image {im_num} ({num_bbox} bounding box(es))
**Question**: {question}
**Thought Process**:
**Answer**:

Figure 13: Prompt for reasoning and answer generation.

**In-Context Learning examples go here**
Similarly to the examples provided above, please imagine this person's thought process behind the areas of the image the person looked at in order to solve the question, which are marked with the red bounding boxes.
The thought process should meet the following conditions:
- It should describe the broader setting of the image and how that context informs the search for specific details.
- It should explain how the bounding boxes (e.g., 'R1', 'R2') were used to narrow down the areas of interest. For instance, it should mention how the image's orientation or the placement of objects guided the focus to certain regions.
- It should only consider the bounding boxes that does appear in the image and not refer to any bounding boxes that actually do not exist. Note that all bounding boxes are numbered sequentially, so if we have 4 bounding boxes in the image, there are exactly the following bounding boxes: 'R1', 'R2', 'R3', and 'R4'.
- It should list specific visual cues noticed within each highlighted region and how these observations lead to identifying the relevant subject (e.g., a man with a backpack).
- It should clearly connect each observation to the final answer. It should explain how, based on the evidence gathered from the image, the conclusion was reached (e.g., identifying the man holding a cell phone).
- If you realize you wrote something wrong in your thought process previously, you should identify and correct the mistakes you have made.

Write as if you are this person, in a similar style as above examples.

## Image {im_num} ({num_bbox} bounding box(es))
**Question**: {question}
**Thought Process**:
**Answer**:

{generation_prompt}

{false_reasoning}

Figure 14: Prompt for self correction. {false_reasoning} is the injected incorrect sub-reasoning.

**Evaluation Task Instructions:**
You will be given three pieces of information:
1. A **Question**
2. A **Model Output** (the answer and rationale generated by a language model)
3. A **Ground Truth Answer** (the correct answer)

Your task is to find model's final conclusion of the question and determine if the final conclusion from **Model Output** is essentially equivalent to the **Ground Truth Answer** and correctly answers the **Question**. Consider the following guidelines when evaluating:
- If the final conclusion includes extra details, omissions, or slight wording differences but the overall meaning and essential information match the ground truth, consider it **Correct**.
- If the final conclusion provides incorrect information, adds unrelated details, or misses critical parts of the ground truth, consider it **Incorrect**.

**Output Format (follow this exactly):**

```

Reasoning: [Provide a brief explanation of why you judged the answer as Correct or Incorrect. Be explicit about which details match or differ.]
Correctness: [Write only one word: Either "Correct" or "Incorrect"]
```

**In-Context Learning examples go here**

**Instance to Evaluate:**
*Question:* {question}
*Model Output:* {model_output}
*Ground Truth:* {ground_truth}

Figure 15: Prompt for long-form answer judgment.

You will be given three pieces of information:
1. A **Question**
2. A **Model Output** (the answer generated by the model)
3. A **Ground Truth Answer** (the correct answer)

**Your task**: Determine if the **Model Output** correctly answers the **Question** by checking whether it aligns or essentially matches the **Ground Truth Answer**, considering meaning rather than exact wording.

Follow these expanded guidelines carefully:

- Mark as **Correct** if the meaning or essential details in the Model Output closely align with the Ground Truth Answer, even if different wording, synonyms, or minor descriptive variations are used.

- **Example**: "Beige," "tan," "cream," or "light brown" are considered close enough to be marked as **Correct**.
- Similarly, minor differences like "couch" vs "sofa," or "silver-colored" vs "metallic grey" should still be considered **Correct** if they describe essentially the same thing.
- Mark as **Incorrect** only if the Model Output significantly differs from the Ground Truth Answer, changing the core meaning or providing substantially different or contradictory information.

- **Example**: If Ground Truth Answer is "wood," and Model Output is "metal," this would be marked **Incorrect** due to clear contradiction.

**Output Format (follow this exactly):**

```

Reasoning: [Provide a brief explanation of why you judged the answer as Correct or Incorrect. Be explicit about which details match or differ.]
Correctness: [Write only one word: Either "Correct" or "Incorrect"]
```

**In-Context Learning examples go here**

**Instance to Evaluate:**
*Question:* {question}
*Model Output:* {model_output}
*Ground Truth:* {ground_truth}

Figure 16: Prompt for short-form answer judgment.

# Task Description:

You will be provided with three distinct pieces of information related to a reasoning task performed by another model:

1. **Initial Reasoning**: An incomplete original response provided by a model containing incorrect reasoning.
2. **Correction**: A concise statement explicitly pointing out the specific mistake or incorrect reasoning in the Initial Reasoning and providing the correct information.
3. **Model's Continued Reasoning**: The model's new attempt at correcting the previously identified incorrect reasoning while complete the initial reasoning.

Your job is to carefully evaluate whether the **Model's Continued Reasoning** successfully identified and corrected the incorrect reasoning based on the provided **Correction**.

—

# Evaluation Steps:

When evaluating, strictly follow these steps:

### Step 1: Identify the Incorrect Reasoning

* Carefully read the **Initial Reasoning** and note exactly what mistake was made based on the provided **Correction**.

### Step 2: Verify the Provided Correction

* Clearly understand the correction, noting specifically which detail from the **Initial Reasoning** was incorrect and what the accurate information is.

### Step 3: Evaluate the Model's Continued Reasoning

* Read the **Model's Continued Reasoning** carefully to verify if the model explicitly identifies the original mistake.
* Ensure the model incorporates the corrected information provided by the **Correction**.
* Check whether the revised reasoning removes the previous mistake and accurately resolves it, not simply ignoring or sidestepping the original error.

### Step 4: Provide Clear Reasoning for Your Judgement

* Explicitly state how the revised model output relates to both the initial incorrect reasoning and the provided correction.
* Clearly explain whether or not the original error was addressed and resolved.

### Step 5: Final Judgement

* Clearly state your final judgement as either **"YES"** or **"NO"**:

* **"YES"** if the revised output explicitly corrects the initial error based on the provided correction.
* **"NO"** if the revised output either does not identify the initial mistake explicitly, fails to correct it, or leaves the incorrect reasoning intact.

—

**In-Context Learning examples go here**
—
# Final Notes for Your Judging:

* Be meticulous and explicit in your reasoning.
* Your evaluation must precisely check whether the revised output aligns correctly with the provided correction.
* Ensure the final judgement ("Yes" or "No") directly and clearly corresponds to your detailed reasoning.

—

**Output Format (follow exactly):**

```
Reasoning: [Provide a clear explanation of your judgment, explicitly mentioning the details or synonyms that align or differ.]
Correctness: [Write exactly one word: Either "Yes" or "No"]
```

—

**Instance to Evaluate:**
*Initial Reasoning:* {false_reasoning}
*Correction:* {model_output}
*Model's Continued Reasoning:* {ground_truth}

Figure 17: Prompt for self-correction judgment.

