# OpenReview forum: "MagiC: Evaluating Multimodal Cognition Toward Grounded Visual Reasoning"
_ICLR.cc/2026/Conference — Submitted to ICLR 2026_

### Official Review · Reviewer_2nqF · 2025-10-23

**Soundness:** 3
**Presentation:** 3
**Contribution:** 3
**Rating:** 6
**Confidence:** 4

**Summary:**

The paper proposed another VQA benchmark dataset. The idea is to enhance grounded multimodal cognition. A set of evaluation metrics were also proposed with the benchmark. The benchmark was evaluated on 15 VLMs and revealed some interesting failure modes of these models.

**Strengths:**

- The benchmark dataset is well designed with broad coverage on different dimensions related to grounded vision cognition.
- The proposed metrics align well with the design target.
- The analysis of model failure modes is informative.

**Weaknesses:**

- There are already many benchmark datasets for VLMs. This one seems to be object-centric with heavy use of bounding boxes. Discussion on the limitations and how it fits into the broader zoo of benchmarks would be helpful.
- The use of LLM judges could introduce biases.

**Questions:**

It would be helpful to provide side-by-side contrasts with existing benchmark datasets.

---

> ### Author Response · Authors · 2025-11-22
> **Clarification/Rebuttal by Author**
>
> Hi Reviewer 2nqF, thank you for the positive assessment of our work, specifically for recognizing the strong design of our benchmark, the alignment of our metrics, and the informative nature of our failure mode analysis. We address your concerns and questions below.  If our response has answered your questions and addressed your concerns, we gently request that you consider raising your score to support the acceptance of this work. Please let us know if there are any further questions that we can answer or clarify.
>
> **1: Distinctiveness from existing benchmarks and "object-centric" nature**
>
> You raised a valid point regarding the saturation of VLM benchmarks. However, we respectfully emphasize that MAGIC fills a critical gap that current benchmarks (like MMBench, MMMU, or MM-Vet) do not address. As illustrated in Figure 1, existing benchmarks primarily evaluate end-task performance (final answer accuracy). They cannot determine if a model arrived at the correct answer for the right reasons or simply exploited dataset biases (hallucination/shortcuts). MAGIC is designed to evaluate the reasoning process itself—specifically, whether the intermediate reasoning steps align with the visual evidence.
>
> The reliance on bounding boxes is intentional. To quantitatively measure grounded cognition, we require precise spatial definitions of visual evidence. While this makes the dataset object-centric, we found it to be the most straightforward and reliable method to verify if a model is "looking" at the relevant evidence during its chain of thought, as opposed to trying to achieve the same with overall scene captioning or attention maps.
>
> We agree that transparency regarding limitations is vital. We have included a dedicated Limitations section in Appendix F, where we explicitly discuss the reliance on GQA and the focus on object-centric scenarios (excluding OCR or abstract math tasks).
>
> **2: Potential Bias of LLM Judges**
>
> We share the concern regarding LLM judges. To mitigate this, we employed a hybrid evaluation strategy designed to minimize bias:
>
> Human-Curated Test Set: Unlike many benchmarks that are fully synthetic, our test set consists of 896 human-curated examples with 5,500 fine-grained annotations (answers, rationales, and bounding boxes) to ensure ground-truth reliability.
>
> Deterministic Grounding Metrics: Our primary novelty, MagiScore, is calculated based on the overlap between predicted regions and human-verified ground truth boxes. This is a numerical calculation, not an LLM judgment, making it completely free of LLM bias.
>
> Conservative LLM Use: We use LLMs primarily for Final Answer Accuracy (to handle synonyms, e.g., "tan" vs. "brown") where exact string matching fails. For reasoning quality, we verified our findings with human evaluation (StepSense, Table 3), which correlates with our automated metrics.
>
> **3: Side-by-side contrasts with existing benchmark datasets**
>
> Per your suggestion, we have summarized the contrast between MAGIC and leading existing benchmarks below:
>
> | Feature             | Existing Benchmarks (e.g., MMBench, MMMU, POPE) | MAGIC (Ours)                                               |
> |---------------------|-------------------------------------------------|------------------------------------------------------------|
> | Primary Goal        | Evaluate Final Answer Accuracy (Success Rate)   | Evaluate Reasoning Process & Visual Grounding              |
> | Input Modality      | Image + Question                                | Image + Question + Bounding Box Proposals (as constraints) |
> | Evaluation Scope    | Holistic / Task-Level                           | Step-by-Step (intermediate reasoning validity)             |
> | Hallucination Check | Binary (Yes/No exist)                           | Spatial (Did the model look at the right place?)           |
> | Diagnostics         | limited to final output                         | Self-Correction & Adversarial Grounding                    |

---

### Official Review · Reviewer_QtJn · 2025-11-01

**Soundness:** 2
**Presentation:** 2
**Contribution:** 2
**Rating:** 2
**Confidence:** 4

**Summary:**

This paper observes that most VQA tasks only evaluate the final answer, while there are few structured evaluations of the reasoning steps themselves. Therefore this paper proposes to enhance part of the GQA dataset with both reasoning steps and visual grounding annotations to be able to explicitly assess these aspects. More specifically, they take 896 questions from the validation set of this benchmark and feed these to four LLMs: InternVL-2.5B, Gema-3-27B. Qwen2.5-VL-7B, and OpenAI GPT4O. This results in reasoning steps which are verified by humans into being True or False. Then the reasoning is corrected in such a way that the reasoning pattern is still valid (e.g. a wrong reasoning step will be followed by a human annotation observing its mistake and correcting it). But I am actually not sure how this would work for multiple wrong reasoning steps in a row. The corrections are used to evaluate the model ability to ‘self-correct’. This is an interesting experiment where they feed the model the question with the reasoning trace up to the mistake, after which they ask the model to complete the rest. But it is also slightly flawed in that it measures an off-policy situation which may be easier to correct for a model.
In addition, they use existing ground truth bounding boxes and divide them into ‘relevant’ and ‘irrelevant’ for the question using a saliency map. The saliency map is obtained through a human which solves these questions (previously done by Chen et al. ECCV2020). The idea is to link these boxes to the reasoning steps to understand whether the model ‘looks’ at the correct parts of the image. Unfortunately, it is not properly explained how (L264 mentions a region extraction function without any details).

Results show that accuracy of a model is correlated with correct visual grounding, which seems unsurprising but the grounding itself seems also very implicit. Next, they show that stronger models are better in (off-policy) self-correction, which is also unsurprising. They also mention a StepSense measure which apparently should measure whether a reasoning step is correct, but this measure is never explained so I am not sure how to interpret this.

**Strengths:**

* It makes sense to evaluate grounding. So collecting reasoning traces makes sense.
* The idea of explicit human corrections to evaluate the self-correction capabilities of a model is fun in theory. However, since it measures correcting off-policy behaviour, it is unclear if it really measures ‘self-healing’ behavior. Models typically tend to be very certain of the reasoning steps which they just produced.

**Weaknesses:**

* There are existing papers like LLM-as-a-judge [Judging LLM-as-a-judge with MT-bench and Chatbot Arena, NeurIPS’23] which analyze reasoning steps fully automatically. While I do believe that using ground truth should be better, we at least want to know how good such an approach is.
* Gemini 2.5 Pro and GPT-5 yield >85% accuracy on GQA, while GQA is noisy. This suggests that GQA is nearly saturated hence I do not believe that this dataset will be relevant for research much longer. It would be much better to collect reasoning traces for more modern and difficult benchmarks.
* The data collection only consists of collecting reasoning annotations. While it could be valuable, by itself the contribution seems rather limited for a top conference like ICLR.
* The insights obtained through the experiments are not very surprising: better grounding leads to better results for a VQA task which looks at relations between objects. Bigger models are better in correcting mistakes (made by other models).
* StepSense remains undefined while the abstract suggests it is an important part of the paper.
* The error analysis is purely qualitative and therefore not very informative.
* Human saliency maps are generally quite noisy. And since there is no quality control of the relevant and irrelevant boxes, this makes it hard to judge the quality of the data.
* More problematically, it remains undefined how regions are mapped to reasoning traces. The qualitative examples seem to suggest that regions are fed in the input by directly drawing on the image in a set-of-marks fashion, but it could also be just a visualization. This is a crucial detail missing from the paper.
* A single extra sentence about what Chen et al. do would prevent requiring the reader to check the AiR-D paper.
* Using LLMs to compare predicted with ground truth responses is not new. Please add some references like [A].
* Not really a weakness, but there is contemporary work in Video QA benchmarks which collects an explicit reasoning trace [Minerva, Nagrani et al. https://arxiv.org/abs/2505.00681]. Please consider citing.
* There is also a portion of synthetically annotated reasoning traces, but these remain unused and it is also unclear whether these would even be useful to the community.
* It would help to have an example of a human correction in L236. Since this is really an important aspect of the paper, it should be clear how these could look like.
* I did not understand the difference between short answer accuracy and long answer accuracy. Especially since the reasoning traces seem to be evaluated in the StepSense metric.
* Table 5.1 is really dense. To make better the point, I would recommend moving the whole table to the appendix. Then I don’t see any significant differences between Full and Short answer correctness, nor between any of the Micro/Macro variants. It would be much easier to present a single scatterplot with Short (or Full) answer correctnes vs the F1 measure of Micro (or Macro) MagiC score.

[A]Jannis Bulian, Christian Buck, Wojciech Gajewski, Benjamin Boerschinger, and Tal Schuster. Tomayto, tomahto.
beyond token-level answer equivalence for question answering evaluation, EMNPL 2022

**Questions:**

Please focus on the major weaknesses.

Please also note that while my review is very critical, I do think the general direction has a lot of potential and I strongly encourage the authors to keep pursuing this direction. But it will require quite a lot of extra work before it reaches the required level for a top-tier conference.

---

> ### Author Response · Authors · 2025-11-22
> **Clarification/Rebuttal by Authors**
>
> Hi Reviewer QtJn, thank you for the detailed comments and suggestions. We will cite the missing works you pointed out as well as add a one sentence description of Chen et al.’s AiR-D work. Below we would like to address your concerns and question you raised in the review.
>
> **1: Clarification on scope, contribution, and relation to prior work**
>
> While we agree that there are existing works using LLM-as-a-judge to evaluate reasoning steps automatically, these works have focused only on the textual domain. MagiC is a multi-modal benchmark mainly focused on diagnosing how existing LVLMs ground their reasoning in image regions, whether their intermediate steps are correct, and whether they can self-correct when confronted with wrong reasoning. As our experiment result showed, LVLMs still struggle to understand the regions of a given image which makes it unsuitable to use as a judge to judge if a given reasoning step is correct w.r.t the region it is focusing on.
>
> In MagiC, GQA is used as a structured source of images and scene graphs, but the benchmark itself is about grounded reasoning rather than answering GQA questions alone. We show that even when models reach high answer accuracy, there is still large headroom in MagiScore and StepSense. This indicates that grounded reasoning on these scenes is far from being saturated.
>
> Regarding contribution, beyond collecting the correctness of each reasoning step, annotators also provide corrections to the incorrect reasoning traces. MagiC also defines a comprehensive evaluation suite (final answer, region focus, step-wise correctness, self-correction), introduces concrete metrics for each component, and evaluates a wide range of LVLMs under a unified protocol. We will highlight that the main contribution is a diagnostic benchmark that exposes where existing LVLMs fail at grounded reasoning, not a new model or a small data add-on.
>
>
> **2: Human saliency quality**
>
> We acknowledge that raw human saliency maps can be noisy, and we did not describe our processing pipeline clearly enough. In MagiC, we use AiR-D eye-tracking maps only as a starting point and apply a dedicated processing pipeline: we threshold and clean the heatmaps, convert them into candidate regions, then co-reference these with GQA-provided object regions. These regions are then being plotted onto each image with different colors for our annotators to validate the relevance w.r.t the question (and correct is necessary). We believe this processing pipeline ensures the quality of our benchmark.
>
> **3. Clarification of the definition for StepSense and methodology**
>
> StepSense is computed from human step-wise annotations over the reasoning chain: (# steps labeled correct) / (total steps). We will add this definition to Sec. 4. Figure 3 in the paper shows a working example of how all the regions are projected to the image before feeding to a LVLM. We will revise Sec 3.1 to clearly define how regions are mapped to reasoning traces. In addition, we would also like to refer reviewers to our Figure 8 (page 14) for an example of a human correction.
>
> **4: Clarification on the weakly supervised (WS) dataset**
>
> Thank you for the request for clarification. The weakly supervised (WS) split is constructed using only GQA object bounding boxes without human saliency and serves as a large-scale training resource. We fine-tuned Qwen-VL 2.5 7B on this WS dataset, and the results show that exposure to WS reasoning traces improves region-focus behavior and overall reasoning coherence, supporting the usefulness of the WS portion of MagiC. Results are shown in the table below and we will also revise our main table to include the numbers.
>
> | Model                     | Size | Final Answer |       | MagiScore (Macro) |           |       | MagiScore (Micro) |        |       |
> |---------------------------|:----:|:------------:|:-----:|:-----------------:|:---------:|:-----:|:-----------------:|:------:|:-----:|
> |                           |      |     Full     | Short |     Precision     |   Recall  |   F1  |     Precision     | Recall |   F1  |
> | QwenVL 2.5                |  7B  |     51.00    | 55.30 |       53.49       |   42.60   | 43.15 |       67.93       |  38.51 | 49.15 |
> |    + SFT                  |      |     58.14    |       |       97.16       |   75.83   | 82.17 |       96.06       |  66.47 | 78.57 |

---

### Official Review · Reviewer_ZPU4 · 2025-11-06

**Soundness:** 2
**Presentation:** 2
**Contribution:** 2
**Rating:** 4
**Confidence:** 4

**Summary:**

This paper introduces MagiC, a new benchmark for evaluating grounded multimodal cognition in large vision-language models (LVLMs). The benchmark combines 5,534 weakly supervised and 896 human-curated question-answer pairs, each annotated with step-by-step reasoning traces and bounding-box groundings. MagiC proposes several evaluation axes beyond final answer accuracy, including (i) reasoning validity, (ii) grounding fidelity, and (iii) self-correction ability. It also introduces new metrics MagiScore for region focus, StepSense for reasoning quality, and Self-Heal for introspective correction. 15 LVLMs (ranging from 7B to 90B parameters, including GPT-4o and Qwen-2.5-VL variants) are systematically evaluated. The authors find that models focusing more precisely on relevant image regions tend to produce better answers, and that larger models show stronger self-correction behavior.

**Strengths:**

* The dataset covers multiple reasoning types with detailed human annotations and saliency-based grounding boxes. The dual-source design (weakly supervised + curated) balances scalability and annotation quality.
* Testing 15 diverse models, including both open and proprietary ones, provides a convincing comparative landscape.
* The addition of StepSense and Self-Heal metrics is creative and useful.
* The paper's main strength lies in its multi dimensional evaluation. The explicit assessment of (1) answer correctness, (2) reasoning validity, (3) visual grounding, and (4) self-correction provides a holistic and insightful view of a model's capabilities than traditional VQA benchmarks.

**Weaknesses:**

* The evaluation results shows the result for a single run, without any standard deviations.
* The dataset is derived from GQA, which limits its diversity, this may not generalize to broader tasks.
* Many metrics rely on a LLM as-a-judge. Without a human agreement study or cross-model validation, it’s unclear how reliable or unbiased these judgements are.
- Formulation of MagiScore seems a bit weak and ambiguous. No concrete details are given on how the various components of this score are computed and how Precision, Recall and F1 of MagiScore relates to standard models of evaluation metrics like attention (please see "Questions" section)
- The formulation of StepSense is not discussed.
- The paper introduces a large, 5,534-example weakly-supervised (WS) dataset, but its role in the evaluation is not fully clarified.
- While the benchmark measures grounded reasoning and attention, “cognition” suggests higher-level abstraction and compositional skills that aren’t really tested here. The framing may be slightly overselling the scope.

**Questions:**

* Section 4.2: What type of prompts are given to the LLM judge?
* Box_q has adversarial boxes as well. So, if the entire reasoning focuses on the adversarial regions, would not the MagiScore still be high?
* What is the region extractor function $\phi$, and how does it map each reasoning step to an index in Box_q ?
* If y_k{i) and y^*{i} is 1, does that mean that the model predicted the exact bounding box as GT or that there is a significant percentage of overlap between these two bounding boxes?
* Section 4.1: IoU measure the overlap between GT and predicted BBoxes. In MagiScore, given a GT BBox and a predicted BBox, how exactly are we checking their similarity?
* In Section 4.3 for Self-heal, how can S_rest = [s_i \dot s_j] semantically match with S_wrong = [s_i \dot s_j ] given i < j? This needs more clarification?
-  How do you define Stepsense?
- Section 5.2: How does the F1 Score of MagiScore relate to attention? Line 371 claims “GEMMA 3 12B, which scored 67.7% in attention” which corresponds to F1 score of MagiScore(micro) is Table 2.
- Could a verbose or generic reasoning chain inflate StepSense?

---

> ### Author Response · Authors · 2025-11-18
> **Clarification/Rebuttal by Authors**
>
> Hi Reviewer ZPU4, thanks for your positive feedback regarding our benchmark and we would like to address the concerns and questions you raised in the review. If our response has answered your questions and addressed your concerns, would you kindly consider raising the score? Please let us know if there are any further questions that we can answer or clarify.
>
> **W1: Single-run evaluation**
>
> Thank you for the observation. Due to limited computational resources, our main experiments report results from a single run per model, which is consistent with prior LVLM benchmark practice.
>
> **W2: Human agreement and cross-model validation**
>
>
> We appreciate the concern. Our benchmark includes human annotations with substantial inter-annotator agreement (Cohen’s κ = 0.72), demonstrating strong consistency among annotators.
>
> **W3: Clarification on the weakly supervised (WS) dataset**
>
> Thank you for the request for clarification. The weakly supervised (WS) split is constructed using only GQA object bounding boxes without human saliency and serves as a large-scale training resource. We fine-tuned Qwen-VL 2.5 7B on this WS dataset, and the results show that exposure to WS reasoning traces improves region-focus behavior and overall reasoning coherence, supporting the usefulness of the WS portion of MagiC. Results are shown in the table below and we will also revise our main table to include the numbers.
>
> | Model                     | Size | Final Answer |       | MagiScore (Macro) |           |       | MagiScore (Micro) |        |       |
> |---------------------------|:----:|:------------:|:-----:|:-----------------:|:---------:|:-----:|:-----------------:|:------:|:-----:|
> |                           |      |     Full     | Short |     Precision     |   Recall  |   F1  |     Precision     | Recall |   F1  |
> | QwenVL 2.5                |  7B  |     51.00    | 55.30 |       53.49       |   42.60   | 43.15 |       67.93       |  38.51 | 49.15 |
> |    + SFT                  |      |     58.14    |       |       97.16       |   75.83   | 82.17 |       96.06       |  66.47 | 78.57 |
>
> **W4: Clarification of MagiScore and StepSense formulation**
>
> For detailed definitions, we refer reviewers to our response to Question 2. Briefly, MagiScore measures region-focus quality via micro- and macro-averaged precision/recall over relevant bounding boxes, while StepSense captures human-evaluated correctness and coherence of intermediate reasoning steps. MagiScore is formally defined in the paper and illustrated in Appendix D and StepSense is explained in out response to Question 7.
>
> **Q1**
>
> The full judge prompts are provided in Figures 15 and 16 (page 19 of the appendix).
>
> **Q2**
>
> Box_q contains all boxes in the image, including adversarial ones. For each box we store a ground-truth relevance flag (relevant vs. adversarial). MagiScore compares boxes mentioned in the reasoning with the set of relevant boxes. Steps focusing on adversarial boxes are counted as irrelevant and lower MagiScore. A worked example is given in Appendix D (page 15).
>
> **Q3**
>
> Reasoning must mention regions in a fixed textual format with box indices (e.g., “Region 1”), enforced during data creation and prompting. Appendix D shows how we parse these references and align them with Box_q.
>
> **Q4**
>
> Equality is over box identities, not IoU. y_k = 1 means the reasoning attends box k; y*_k = 1 means box k is labeled relevant. Both being 1 means the model attended the same predefined box index as the ground truth. This is possible because we plot predefined boxes and have the model to select among them instead of predicting coordinates.
>
> **Q5**
>
> In MagiScore, we do not ask the VLM to pinpoint bounding box coordinates. We (1) predefine and plot all regions and (2) let the model select boxes by index in its reasoning. We then check similarity by index match with ground-truth relevant boxes. This avoids localization errors seen in our early experiments and makes sure this benchmark focuses on grounded reasoning rather than precise coordinate pinpointing/IoU.
>
> **Q6**
>
> S_rest = [s_{j+1}, s_{j+2}, …]; it starts from j+1 (not i), so S_wrong and S_rest do not overlap.
>
> **Q7**
>
> StepSense is computed from human step-wise annotations over the reasoning chain: (# steps labeled correct) / (total steps). We will add this definition to Sec. 4.
>
> **Q8**
>
> In that sentence, “attention” informally meant MagiScore(micro). To avoid confusion, we will revise Sec. 5.2 to say “MagiScore” explicitly.
>
> **Q9**
>
> StepSense requires human annotation for calculation, and we instruct annotators to flag generic, repetitive steps, which are discarded, so they cannot boost the score. Annotators also reported that reasoning was not generic, mostly due to the in context learning example provided in the prompt.
>
>
> Again, we appreciate all your suggestions to strengthen this work!

---

### Meta-Review · Area_Chair_Nbcy · 2026-01-02

**Summary:**

Paper #13695 introduces a benchmark for multimodal reasoning based on the GQA dataset and additional annotations of reasoning steps. It received three reviews with ratings 2, 4, 6. The only (slightly) positive review, providing a rating of 6, was extremely short and did not have much impact on the reviewing process.

The reviewers appreciated the general idea of annotating reasoning steps and to correct the models' reasoning and partially feed it back to the models, evaluating "self-correction", although some questions of the technical soundness were raised. The main weaknesses were
- depth/breath of the contributions,
- Positioning wrt the SoTA, in particular papers also evaluating reasoning steps,
- GQA as a very limited choice of base dataset,
- Unsurprising insights and non informative performance analyses,
- Missing details, even up to the missing definition of a newly introduced metric, and other technical details.

The authors attempted to answer some of the weaknesses, but critical issues remained: while the idea is very interesting, its execution lacks depth and in fine the limiting analysis does not lead to new insights for the field. The AC judges that the paper is not yet ready.

**Reviewer Concerns:**

See above.

**Reviewer Scores:**

No information available.

---

### Decision · Program_Chairs · 2026-01-26

Reject